# Automatic Detection of Feral Pigeons in Urban Environments Using Deep Learning

**DOI:** 10.3390/ani14010159

**Published:** 2024-01-03

**Authors:** Zhaojin Guo, Zheng He, Li Lyu, Axiu Mao, Endai Huang, Kai Liu

**Affiliations:** 1Department of Infectious Diseases and Public Health, City University of Hong Kong, Hong Kong SAR, China; zggguo2-c@my.cityu.edu.hk (Z.G.); axmao2-c@my.cityu.edu.hk (A.M.); 2School of Communication Engineering, Hangzhou Dianzi University, Hangzhou 310018, China; 3Department of Computer Science, City University of Hong Kong, Hong Kong SAR, China; edhuang2-c@my.cityu.edu.hk

**Keywords:** wildlife survey, urban ecosystems, animal welfare, computer vision, automatic counting

## Abstract

**Simple Summary:**

We advanced a deep learning model that significantly enhances the detection and population estimation of feral pigeons in the dynamic urban landscape of Hong Kong, employing computer vision techniques. The inherent challenges associated with pigeon concealment within complex urban structures and their high mobility necessitate a robust and effective strategy. Our improved model, Swin-Mask R-CNN with SAHI, integrates a Swin Transformer network for deep feature extraction, a feature pyramid network to enhance multi-scale learning, and three distinct detection heads for classification, bounding box prediction, and segmentation of feral pigeons, respectively. With the assistance of the Slicing-Aided Hyper Inference tool (SAHI), our model excels at detecting small-target pigeons in high-resolution images. Experimental results have demonstrated a substantial 10% increase in AP_50s_ (average precision at 50% intersection over union) compared to the Mask R-CNN approach. This improvement signifies the immense potential of our model in dynamic pigeon detection and accurate population estimation. The success of our novel approach provides a promising solution for effectively managing urban wildlife populations.

**Abstract:**

The overpopulation of feral pigeons in Hong Kong has significantly disrupted the urban ecosystem, highlighting the urgent need for effective strategies to control their population. In general, control measures should be implemented and re-evaluated periodically following accurate estimations of the feral pigeon population in the concerned regions, which, however, is very difficult in urban environments due to the concealment and mobility of pigeons within complex building structures. With the advances in deep learning, computer vision can be a promising tool for pigeon monitoring and population estimation but has not been well investigated so far. Therefore, we propose an improved deep learning model (Swin-Mask R-CNN with SAHI) for feral pigeon detection. Our model consists of three parts. Firstly, the Swin Transformer network (STN) extracts deep feature information. Secondly, the Feature Pyramid Network (FPN) fuses multi-scale features to learn at different scales. Lastly, the model’s three head branches are responsible for classification, best bounding box prediction, and segmentation. During the prediction phase, we utilize a Slicing-Aided Hyper Inference (SAHI) tool to focus on the feature information of small feral pigeon targets. Experiments were conducted on a feral pigeon dataset to evaluate model performance. The results reveal that our model achieves excellent recognition performance for feral pigeons.

## 1. Introduction

The overpopulation of feral pigeons can lead to an imbalance of environmental and human health in the urban ecosystem. Excessive droppings from dense pigeon populations contaminate air and water resources, while these birds can harbor pathogens like chlamydiosis and cryptococcosis, transmissible to humans via respiratory secretions, feathers, and feces [1], thereby elevating infection risks for vulnerable individuals. In addition, overpopulation will negatively affect urban infrastructure due to increased excess of feces. Then, the feces of feral pigeons usually damage valuable buildings and statues and cause a huge economic loss [2]. Consequently, monitoring and quantifying feral pigeon populations is essential for evaluating their distribution and identifying instances of overpopulation, ultimately informing the design of effective intervention strategies. Traditional studies of feral pigeons typically utilize the mark–recapture method [3], point–count surveys [4], and nest-site surveys [5] to record the number of pigeons. These methods are labor-intensive and inaccurate due to observation difficulties. Moreover, traditional counting methods of feral pigeons present a formidable challenge due to factors such as object fast movement, overlap, obscured visibility, and varying population density across environments.

Deep learning [6] has rapidly emerged as a potent and efficient solution for object detection [7] and counting across diverse settings, replacing traditional manual methods. Its applications apply to the realm of animal detection [8,9] and counting. Some deep learning approaches address animal detection from a single scene, such as monitoring pigs, sheep, and cattle [10,11,12,13]. However, bird detection [14] scenarios present greater complexity and variability in target scales compared to the relatively uniform environments of livestock detection and counting. 

To solve the problem of bird detection and counting, a series of enhanced deep learning techniques have been employed. In its early attempts, the CNN algorithm showed promise in boosting bird detection accuracy [15]. For expedited bird detection, the You Only Look Once (YOLO) strategy [16], which adopts a one-stage approach, has been utilized. A model dubbed DC-YOLO, based on YOLOv3, was devised to accurately detect bird populations near power lines [17], and one modified YOLO model was used to detect cage-free hens on the litter floor [18]. Furthermore, a temporal boosted YOLO model was constructed for detecting birds in specific wind farms [19], and a combination of YOLO and Kalman filter [20] was applied to low-light images for detecting and tracking chickens [21]. These methods have improved bird detection accuracy. However, the modified YOLO algorithms’ key limitation is their fast one-time detection, making frequent loss of some objects. Compared with one-stage detection models [22], two-stage networks [23] place greater emphasis on image details. Hong et al. [24] developed a two-stage deep learning-based bird detection model for monitoring avian habitats and population sizes. Their study utilized a dataset containing diverse bird habitats, lakes, and agricultural landscapes and compared the performance of YOLO, Faster R-CNN [25], and SSD [26] detection models. The results indicated that the Faster R-CNN model exhibited great detection accuracy. To further augment image detail extraction capabilities, two-stage Mask R-CNN [27] was introduced to improve object detection accuracy. Nevertheless, ample room remains for enhancing the detection capabilities of deep learning algorithms for bird detection.

To the best of our knowledge, no extant research presents a feral pigeon detection model suitable for complex urban environments. In pursuit of this objective, we propose a Swin-Mask R-CNN with SAHI model for feral pigeon detection. First, this model adopts the Swin Transformer [28] as its backbone network, utilizing a hierarchical local attention mechanism and cross-layer information exchange to decrease computation and enhance feature extraction capabilities, respectively. Second, we select FPN [29] as the neck in this model to merge feature maps of varying scales, thereby improving the model’s detection capacity for differently sized feral pigeon targets. Lastly, we employ Slicing-Aided Hyper Inference (SAHI) [30] for image slicing and feature information allocation for large and small targets. We subsequently apply the Non-Maximum Suppression (NMS) [31] algorithm to identify the optimal detection box and freeze the target segmentation branch during the inference and prediction stages to expedite target detection.

In summary, the proposed Swin-Mask R-CNN with SAHI model further refines the accuracy of small-target object detection and exhibits enhanced generalization capabilities. These advancements hold significant potential for the detection and counting of feral pigeons across diverse urban environments. As far as we are aware, our research represents the first urban pigeon detection and counting initiative. Our major contributions can be summarized as follows:

(1)We created a unique dataset of feral pigeons in an urban environment with manually annotated bounding boxes, concentrating on the detection and enumeration of urban pigeons across diverse cityscapes.(2)We developed Swin-Mask R-CNN with SAHI model for pigeon detection, incorporating the SAHI tool to preserve fine details of smaller targets during the inference phase, thereby enhancing detection accuracy. (3)We further improved the model with SAHI tool and enabled it to encompass a greater number of feral pigeons by utilizing large-scale images (4032 × 3024), achieving broader detection coverage. 

## 2. Materials and Methods

### 2.1. Image Data Collection

Feral pigeon images used in this study were collected from various areas across Hong Kong SAR, China. Two cameras were used for image collection: a main camera with a 12MP Sony IMX503 sensor featuring optical stabilization and HDR mode and a secondary camera with a 12MP Sony IMX372 sensor featuring a 120° ultra-wide-angle function and PDAF phase detection technology. To ensure that the data collected cover different urban environments and pigeon poses, data collectors conducted the following enrichment shots: (1) Different urban environments include park grounds, flowerbeds, tree groups, residential buildings, and sky; (2) Different illumination levels in the dawn, morning, and afternoon; and (3) Different postures of birds include flying (wing flapping), standing, eating, and walking. To avoid disturbing feral pigeons’ normal activities, photos were taken quietly from at least 3 m away. Since feral pigeons are small targets, and in some cases, they may also be far away from the lens or even obstructed by other objects, high-resolution images (i.e., 4032 × 3024 pixels) were collected to obtain a more straightforward target display effect. Consequently, the dataset consists of 400 images, which have diverse backgrounds and pigeon poses. Our dataset was selected with a relatively homogeneous sampling of different urban scenes. Examples of the images are shown in Figure 1, where feral pigeons inhabit various poses at different locations within urban environments. 

### 2.2. Data Labeling and Augmentation

The LabelImg annotation tool [32] was used to label all feral pigeons in the images. For example, the red boxes are shown in Figure 1. Full-body labeling of rectangle boxes was used for fully visible feral pigeons, while only the visible parts were labeled for partially visible feral pigeons (Figure 1). A total of 400 high-resolution images were manually labeled to ensure accuracy, with the associated label information meticulously stored in the standard COCO format JSON file; for this project, to facilitate the model’s learning and recognition of feral pigeon features. Within the realm of target detection, the classification of objects based on size is critical. Specifically, “small” objects are defined as those whose bounding box area ranges from 0 to 1024 pixels (32 × 32), and “medium” objects are those with bounding box areas spanning from 1024 to 9216 pixels (32 × 32 to 96 × 96). By adhering to these criteria, feral pigeons typically fall into the small to medium-sized category when the images are captured from a distance of at least three meters.

Building on the foundation of the initial dataset, we embarked on a rigorous data augmentation process to enhance the robustness of our model. This process is essential to help prevent overfitting and improve the model’s ability. However, the direct application of traditional data enhancement operations on feral pigeon images, like image flipping, image rotation, etc., may inadvertently distort the natural shape of the pigeon, making it impossible to accurately localize the precise position of the feral pigeon in its original environment. Therefore, we tailored our augmentation strategies to preserve the integrity of the pigeons’ shape, ensuring that the model’s training would benefit from both the diversity of the augmented data and the fidelity of the target representations. To effectively perform data enhancement, we use the slicing image of the SAHI tool [30]. The original image (4032 × 3024 pixels) is batch-partitioned into nine sub-images (1344 × 1008 pixels) totaling 3600, after which the sub-images are enlarged to the same size as the original image (4032 × 3024 pixels). It is worth noting that although a few of the segmented images do not contain feral pigeon targets, these images can be used as negative samples to help the model learn how to distinguish between targets and backgrounds. The original dataset was randomly divided into a training set, a testing set, and a validation set in a ratio of 4:1:1. The 400 original images were expanded to 4000 images for training, evaluation, and testing to increase the network’s robustness and prevent overfitting. 

During the image slicing stage, the bounding box information was also converted to correspond to the feral pigeon target in each slice while performing image augmentation. However, scaling may cause smaller targets to become even smaller, and feral pigeons cropped at the segmentation line may be split into different parts, leading to the loss of target-related pixel information. These operations may cause the model to learn feral pigeon features incorrectly during the training. To mitigate these issues, the following methods were used in this experiment: during the scaling process, original boxes that were too small were directly removed, and during the segmentation process, labeling information where the bounding box of the sub-image is significantly smaller than the bounding box of the original image was deleted. The results are shown in Figure 2. After processing all the images and labeling information, the final training and test sets contained 5612 and 1238 feral pigeon objects, respectively. The dataset splitting is shown in Table 1. 

### 2.3. Workflow Overview

The overall workflow of our method is illustrated in Figure 3. The methods described in Section 2.1 were used to construct the dataset in the model training stage. The dataset was then fed into a modified Mask R-CNN network, which was built using Swin Transformer as the backbone, to train the model and achieve high accuracy. SAHI [30] was used in the prediction stage to identify feral pigeons effectively, especially the small ones, and those partially occluded or in shadowed areas. Below are the steps taken in this experiment for feral pigeon target detection.

We constructed an object detection model for feral pigeons using a modified Mask R-CNN network workflow (Figure 4). The model was trained with a dataset of annotated images in SAHI, which were cleaned and augmented to ensure high quality and diversity. After training the model, we extract high-dimension features by the SAHI [30] tool to predict the feral pigeon targets in the prediction process, including the detection of small targets, partially occluded targets, and targets in shadowed areas. Meanwhile, the mask head is frozen to speed up the inference process. The output of the model includes prediction images, which display the predicted results and the position information of the target boxes. In the final process, to assess the effectiveness of the model, the mean average precision (mAP) is utilized as a metric for comparison with other models (i.e., YOLOv5-s, YOLOv5-m, Faster R-CNN, Mask R-CNN) and determine how well it performs in terms of accuracy and robustness. The detailed implementation of the above steps can be found in Section 2.4. The flowchart can be referred to in Figure 4.

### 2.4. Swin-Mask R-CNN with SAHI Model

In this experiment, the images are large-sized images of 4032 × 3024 pixels, and most of the detected objects are small-sized. Therefore, to fully extract the image detail features, the Swin Transformer was used as the backbone network (a), and FPN was used as the improved version of the neck to construct a state-of-the-art modified Mask R-CNN (Figure 4).

In the first stage, the Swin Transformer network is used to extract hierarchical multi-scale image information, and four stages are used to construct four scales of feature layers. Then, FPN is used to fuse large-scale low-level features and small-scale high-level features through upsampling and downsampling to obtain four scales of feature layers with richer information. The feature maps (b) are input to the RPN network (c) to generate candidate regions of different sizes, and the candidate boxes are preliminarily screened. In the second stage, RoI Align (d) is performed on the candidate regions generated in the previous step to extract fixed-size feature maps, and then classification, bounding box regression, and mask regression tasks are performed to obtain more accurate detection boxes in the head module (e).

Finally, in the Swin-Mask R-CNN with SAHI model, the mask head is modified because the prediction stage only involves object detection tasks. We use conditional judgment to ignore the mask head sub-network and only output the two sub-networks of object classification and bounding box regression in the head to accelerate the process of generating images during object detection in the prediction stage.

### 2.5. Swin Transformer Backbone

To construct a multi-scale hierarchical structure [33] for pigeon detection, the Swin Transformer [28] modifies the image dimensions using different operation combinations at various stages. First, 4032 × 3024 RGB images are batched as input to the network, and the images are divided into patches with a patch size of 4 × 4. In Figure 5, the patch partition module then partitions the input images into small regions to expand the network’s receptive field and enhance its feature representation capabilities. The image’s height (H) and width (W) are reduced to a quarter of their original size, while the number of channels is set to 48, resulting in image dimensions of 1008 × 756 × 48. Next, in Stage 1, a Linear Embedding operation [34] is applied to change the vector dimensions to a pre-set value of C = 96. The current H and W dimensions are flattened and stored as a linear dimension, with a sequence length of 762,048. Since this sequence length is too long, a window-based self-attention computation is used in the Swin Transformer block to reduce the sequence length, effectively reducing the complexity of training, and resulting in image dimensions of 1008 × 756 × 96. Following this, in Stage 2, the patch merging method is employed to combine adjacent small patches into larger patches, achieving a similar effect to convolution and providing a downsampling effect for the basic patches. After passing through the Swin Transformer block, the final image dimensions are changed to 504 × 378 × 192. In Stages 3 and 4, the patch merging [28] and Swin Transformer block operations from Stage 2 are repeated, further reducing the image dimensions to 252 × 189 × 384 and 126 × 94 × 768, respectively. Lastly, the image information from the final three channels will be further utilized in the subsequent Feature Pyramid Network (FPN) [28].

The patch merging operation, similar to the pooling operation in convolutional neural networks [35], gradually extracts higher-level abstract features from the original pixel-level features, thereby improving the performance of object detection and classification tasks. In Figure 6, downsampling by a factor of two is first performed, and each basic patch labeled with the numbers 1, 2, 3, and 4 is combined. By performing stride sampling for points with the same index, basic patches with the same label are merged into a larger patch, which helps to construct multi-scale representations and simultaneously increases the network’s receptive field. This operation reduces the image’s height and width dimensions by half. Subsequently, the downsampled image information is concatenated in the channel dimension, resulting in a fourfold increase in the number of channels (C). To achieve the effect of doubling C as in the dimensionality reduction methods used in convolutional neural networks, a 1 × 1 convolution is employed to change the number of channels to 2c. Through these steps, the spatial dimensions of the image width (W) and height (H) are reduced by half, while C is doubled.

### 2.6. Feature Pyramid Network (FPN)

FPN [28] is a robust object detection strategy that merges multi-scale features to handle varying object sizes in images, compared to conventional methods [35]. It includes two modules: a down-to-top feature pyramid construction [36] and a top-to-down feature fusion [37], producing a high-resolution feature map rich in deep semantic information.

In Figure 7, the first module constructs a feature pyramid in a down-to-top manner, grouping feature maps of the same size into stages during the image’s forward propagation through the Swin Transformer backbone network. This process involves convolution, pooling, and activation operations, with the feature map size decreasing from bottom to top. In the second module, the top-to-down feature fusion, the small-scale feature maps containing deeper semantic information are upsampled following the creation of a feature pyramid with decreasing scales in the backbone network. These are then concatenated with the corresponding size feature map from the previous stage, resulting in a high-resolution feature map containing profound semantic information.

### 2.7. Slicing-Aided Hyper Inference (SAHI) Tool

The predictive approach of Swin-Mask R-CNN paired with SAHI is designed to mitigate pixel information loss for feral pigeons, all without requiring additional training. In the conducted experiment, we utilized the SAHI method [30], specifically utilized for small-target feral pigeon detection, to enhance the accuracy of identifying small objects.

In Figure 8, part A presents a direct full inference from the original image. Meanwhile, part B illustrates a process where the original image is divided into nine sub-images. These resulting patches are resized to match the original dimension of 4032 × 3024 pixels, and subsequently fed individually into the Swin-Mask R-CNN model for independent inference. Upon completing a batch of ten images, the detection boxes for each image are computed. The original image serves to identify larger objects, while the nine sub-images assist in enhancing the detection of smaller objects. In part C, all the processed bounding boxes are consolidated. Overlapping predicted targets are managed using Non-Maximum Suppression (NMS) [31]. Specifically, for small and densely packed feral pigeon targets, overlapping boxes often represent different parts of the same target. When the Intersection over Union (IoU) [31] value surpasses a pre-set threshold, the box with the highest confidence score is chosen as the result. Boxes with detection scores falling below the threshold are discarded, thereby refining the detection accuracy. Finally, the remaining bounding boxes, representing the detection results for feral pigeons, are illustrated in part D of Figure 8.

## 3. Results

### 3.1. Experimental Settings and Model Evaluation Indicators

To ensure adequate model fitting, the model underwent training for 200 epochs. The AdamW optimizer was employed with a beta parameter ranging from 0.9 to 0.999. Gradient clipping was not employed. The learning rate was set to 0.0001. Additionally, a step adjustment strategy was utilized, accompanied by a linear warm-up strategy for the learning rate. The warm-up phase consisted of 500 iterations with a warm-up ratio of 0.001. Throughout the training process, the model was saved every two epochs, with the current best model being consistently preserved. We used a GPU for model training and set the batch size to 16. The parameters remained the same for the other models used for comparison. All experiments were conducted on Ubuntu 18.04, and the hardware parameter settings for training, testing, and prediction are shown in Table 2.

Metrics including mean Average Precision (mAP), AP_50_, AP_50s_, and AP_50s_ were used to evaluate the model’s performance. mAP is the average precision score of the model on all categories and is a comprehensive evaluation metric. The mAP formula is given in Equation (1). 

In Equation (1), Precision (p): the proportion of samples that are positive among all samples that are positive in the detection result, where TP is True Positives and FP is False Positives, and the calculation formula is shown in Equation (2). Recall (r): the proportion of samples that are correctly detected as positive among all samples that are positive, where FN is False Negatives, and the formula is shown in Equation (3).
(1)$$mAP=\frac{\sum_{i=0}^{N-1}\int_{0}^{1}\mathrm{PR}dR}{N}$$
(2)$$P=\frac{\mathrm{TP}}{\mathrm{TP}+\mathrm{FP}}$$
(3)$$R=\frac{\mathrm{TP}}{\mathrm{TP}+\mathrm{FN}}$$

AP_50_ is the average precision score when the Intersection over Union (IoU) is greater than or equal to 0.50 and is an evaluation metric for larger targets. AP_50s_ is the average precision score when the IoU is greater than or equal to 0.50 for small targets (areas less than or equal to 32 × 32 pixels).

### 3.2. mAP Comparison of Different Model

Our methodology is evaluated in comparison with one-stage and two-stage object detection algorithms with original and latest backbones under identical experimental conditions [38,39]. We selected two versions of YOLOv5, YOLOv5-s with GhostnetV2 [40], YOLOv5-s with MobilenetV3 [41], Faster R-CNN, Mask R-CNN, Mask R-CNN with MobileViT [42] for performing comparative experiments. In this context, model parameters, mAP, AP_50_, and AP_50s_ served as the evaluation metrics. 

As Table 3 indicates, the two-stage series network, represented by Faster R-CNN and Mask R-CNN, outperformed the one-stage series network (YOLOv5) by achieving a mAP above 50%. YOLOv5-s with GhostnetV2 and YOLOv5-s with MobilenetV3 models did not significantly improve the accuracy of feral pigeons. Moreover, the Transformer-based lightweight network Mask R-CNN with MobileViT can effectively improve the performance of the model with 61% mAP, and by employing Swin Transformer and FPN as the core network within Mask R-CNN to learn more intricate information, the detection accuracy reached a new benchmark with a mAP of 0.68.

Remarkably, the modified Mask R-CNN model’s capacity for small-target recognition was also significantly enhanced, achieving an AP_50s_ of 0.57. This improvement can be attributed to the effective integration of Swin Transformer and FPN, which facilitates better interaction of information from each feature layer, thereby enabling more precise global and local recognition. In conclusion, the use of Swin Transformer and FPN as the backbone of Mask R-CNN within a two-stage network aligns with the ability to boost detection accuracy, particularly with small-target detection.

To initially validate the individual contribution of each component within the modified Mask R-CNN to the model’s overall performance, we conducted controlled experiments under the same conditions using the feral pigeon dataset. In Step 1, we established a baseline model that utilized Mask R-CNN with a state-of-the-art (SOTA) Swin Transformer as the backbone and a Feature Pyramid Network (FPN) as the neck. Subsequent experimental steps involved ablation studies of these components. In Step 2, we began by removing the FPN component to evaluate its contribution. This resulted in a decrease in the model’s mAP, AP50, and AP50 small object scores by 0.04, 0.1, and 0.05, respectively, compared to the baseline. Moving to Step 3, based on the modified model from Step 2, we replaced the sole Swin Transformer backbone with a more generic Resnet backbone. Relative to Step 2, this change further reduced the model’s mAP, AP_50_, and AP_50s_ to their lowest values of 0.51, 0.75, and 0.40, respectively. The results of these ablation experiments demonstrate that both the FPN and the Swin Transformer backbone significantly enhance the accuracy of the model in detecting feral pigeons, compared to the original Mask R-CNN SOTA, detailed results can be found in Table 4.

To further verify that the effectiveness of SAHI tools can improve the accuracy of small-target detection, we incorporated them into all previously mentioned models during inference. The final row in Table 5 indicates that our proposed SAHI, when used in conjunction with Swin Transformer and FPN as the backbone network, yields the most substantial improvement in small-target recognition, achieving an AP_50s_ of 0.66. This surpasses all other models combined with SAHI.

When SAHI is added to YOLOv5-s, YOLOv5-m, YOLOv5-s with GhostnetV2, YOLOv5-s with MobilenetV3, Faster R-CNN, Mask R-CNN, Mask R-CNN with MobileViT, there is a notable increase in the mAP, with scores of 0.51, 0.56, 0.49, 0.53, 0.60, 0.62, and 0.69, respectively. In conclusion, the SAHI tool, by optimizing model recognition outcomes through a greater focus on image details while preserving original results, can improve the detection capability of all models involved in the experiment.

To validate the effectiveness of augmenting the dataset in improving the model’s performance, we conducted experiments using the original dataset, an augmented dataset with an additional 2000 images, and a fully augmented dataset in our Swin-Mask R-CNN with SAHI model. The experimental results showed that augmenting the dataset further enhanced the accuracy of the model in detecting different sizes of feral pigeon objects. Compared to the original dataset, the mAP, AP_50_, and AP_50s_ improved by 0.23, 0.22, and 0.25, respectively, by a fully augmented dataset. Detailed data results can be found in Table 6.

### 3.3. Results Visualization

From the above series of experiments, it is evident that our proposed method greatly outperforms other methods in terms of accuracy. Subsequently, in this section, we present the image results obtained from our model’s inference in comparison to other models and demonstrate our model’s robustness against interference and its proficient recognition ability under density targets of various posture feral pigeons.

In urban environments, feral pigeons are often found with sparrows. Consequently, it is crucial for the model to accurately distinguish between these species and effectively eliminate sparrow interference for accurate identification. As depicted in Figure 9, our model successfully discriminates between feral pigeons and sparrows, thereby preventing erroneous detections of sparrows. 

Figure 10a shows the result of using YOLOv5-s pre-trained weights with the public COCO80 classes for image inference and feral pigeon detection without fine-tuning. Besides predicting other classes such as buses, we can see that the public model also incorrectly predicted the feral pigeon as a person category. Moreover, for the pigeon prediction, the model only had low confidence in predicting the selected object as a feral pigeon. Figure 10b shows the result of using our proposed model for the same inference. Our model can make high-confidence predictions for partially occluded and shadowed feral pigeon targets.

As shown in Figure 11a, the left image shows the original Mask R-CNN model, which still has missed detections for some small targets in large images. Figure 11b clearly demonstrates that our proposed model can easily detect all feral pigeon targets.

Figure 12 shows the dense feral pigeons flying in the sky or standing on the ground we captured. Although feral pigeons have significant differences in their postures, our model can still capture different postures of feral pigeons in different scenes, further demonstrating the robustness of our model.

### 3.4. Pigeon Counting Demo

To achieve dynamic counting of feral pigeons, the current statistical method for feral pigeons allows for the selection of videos of varying durations for analysis. In this study, we selected a 20 s video of feral pigeons and extracted one frame per second from the video stream. Each frame was input into our model for feral pigeon detection, and the resulting image is displayed in Figure 13b. The total number of detected feral pigeons will be shown in the upper left corner of the image, while the dynamic count of feral pigeons will be displayed in Figure 13a after the image inference is completed, and the results indicate that there were 17 feral pigeons present in the 20th second.

## 4. Discussion

To address the challenge of accurately detecting feral pigeons in complex urban environments, we propose an improved Mask R-CNN model called Swin-Mask R-CNN with SAHI Model. To validate the performance of our proposed model, we compare it with other classic detection models. Through researching the former study and two stages of experimental comparisons, we draw the following conclusions:The use of deep learning network algorithms for bird recognition has consistently demonstrated strong performance [43,44]. However, there are several limitations in current bird detection methods. Firstly, the utilization of traditional backbone networks in two-stage detection approaches [45,46] hinders the maximization of network performance in bird detection. Secondly, most existing studies on bird detection using deep learning techniques lack a specific focus on individual bird species, such as feral pigeons. Some studies concentrate on the accurate identification of various bird species in airborne scenarios [14,15,24,46], while others explore the classification and detection of different bird species in natural environments, such as wind farms or aquatic habitats [19,45]. Additionally, a few studies specifically investigate the detection and counting of different bird species in specific regions, such as birds on transmission lines [18]. Moreover, extensive resources are required for traditional feral pigeon research in urban environments [47], and the limited urban pigeon detection focuses only on specific areas such as buildings [48]. There is currently no comprehensive study on feral pigeon detection in complex urban settings. To address these challenges, we propose an automatic detection method for feral pigeons in urban environments using deep learning. Through a series of experiments, we demonstrate the effectiveness of our proposed method in feral pigeon detection in urban areas.In bird detection, most studies have utilized one-stage (YOLO) [15,17,19,21] and two-stage (Faster R-CNN and Mask R-CNN) [43,44] object detection models. The original Mask R-CNN has demonstrated great performance in bird detection [37]. Based on this, we propose an improved algorithm that enhances the main components of the original Mask R-CNN and incorporates the SAHI tool to improve the model’s detection performance. Recent studies have shown the effectiveness of the Swin Transformer in capturing fine-grained animal details [49,50]. Therefore, we replace the backbone of the original Mask R-CNN with the Swin Transformer and add FPN as the network’s neck for multi-scale feature fusion [51]. After adjusting the network, to evaluate the performance of the Swin-Mask R-CNN model, we compare it with commonly used object detection methods for bird detection, including YOLO [16,18,19,21], Faster R-CNN [45], Mask R-CNN [46], and our proposed method (Swin-Mask R-CNN) on our feral pigeon dataset. The mAP of our proposed Swin-Mask R-CNN model reaches the highest value of 0.68. These experimental results demonstrate that by applying various bird detection models and the Swin-Mask R-CNN model to feral pigeon detection, our model achieves the best performance. Moreover, although using Swin-Mask R-CNN as the architecture yields optimal results in the previous comparative experiments, there is still room for improvement in detecting small objects of birds (AP_50s_). There are specific studies focused on the detection of small objects of birds [15,46]. Therefore, to further enhance the accuracy of detecting small objects of feral pigeons, we introduce the SAHI tool [30] to assist inference processing. In this phase, we incorporate the SAHI tool into all the models involved in the previous experiments and conduct further experiments on our dataset. The experimental results demonstrate that our Swin-Mask R-CNN with SAHI model significantly improves the accuracy of feral pigeon detection, achieving the highest values in mAP, AP_50_, and AP_50s_ with improvements of 6%, 6%, and 10%, respectively.The detection and estimation of feral pigeon populations can provide us with a better understanding of their growth and distribution in different urban areas. If there is an occurrence of feral pigeon overpopulation, relevant authorities can take appropriate management measures to prevent their negative impact on the urban environment and avoid excessive competition with other urban bird species, which could disrupt ecological balance and species diversity. Our current work has significantly improved the detection capability of feral pigeons in urban environments, but we still face some challenges in the future. Our research has the following two limitations: we have not further tested the generalization ability of our model, and we have not fully deployed it in real time on portable terminals. In future work, we plan to enhance these aspects. On one hand, although our proposed model demonstrates good detection performance, to further validate its generalization ability, we intend to collect larger datasets encompassing feral pigeons and other bird species from various cities through collaborations with researchers and public data sources. On the other hand, while we have developed a demo for automatic feral pigeon counting, it has not been extensively deployed in real-world scenarios. Our goal for future work is to deploy our algorithm on cloud and mobile platforms, enabling researchers to upload photos and videos for automatic analysis by the model. This will provide feral pigeon detection and counting results, allowing estimation of feral pigeon populations in different areas and assessment of the impact of feral pigeon overpopulation.

## 5. Conclusions

In this study, we introduce a novel modified Mask R-CNN model called Swin-Mask R-CNN with SAHI for feral pigeon detection in Hong Kong urban environments, which aims to detect feral pigeons in large-size images with 4032 × 3024 resolution. Our model uses the Swin Transformer backbone network and FPN to construct a feature map with more detailed information. To capture more small pigeon targets, the SAHI tool is applied to zoom in on the pigeon information, and we finally freeze the segmentation network to speed up the detection process during the inference part. The results demonstrate that Swin-Mask R-CNN with SAHI model architecture has the greater performance for pigeon detection with 74% mAP. Compared with other models, our model can achieve the best detection of feral pigeons in different environments such as bushes, buildings, and cities under the sky. It can identify overlapping pigeons, pigeons in the shadow, flying pigeons, pigeons eating, and walking pigeons.

## Figures and Tables

**Figure 1 animals-14-00159-f001:**
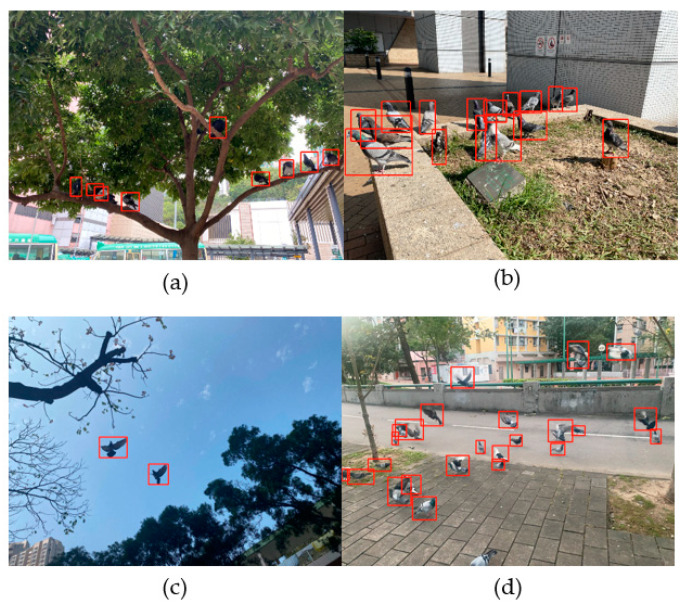
Examples of annotated images of dense feral pigeons inhabiting various poses at different locations within urban environments: (**a**) feral pigeons perching on trees; (**b**) feral pigeons standing in flower beds; (**c**) feral pigeons flying under the sky; and (**d**) feral pigeons flying, standing, and hovering on the street.

**Figure 2 animals-14-00159-f002:**
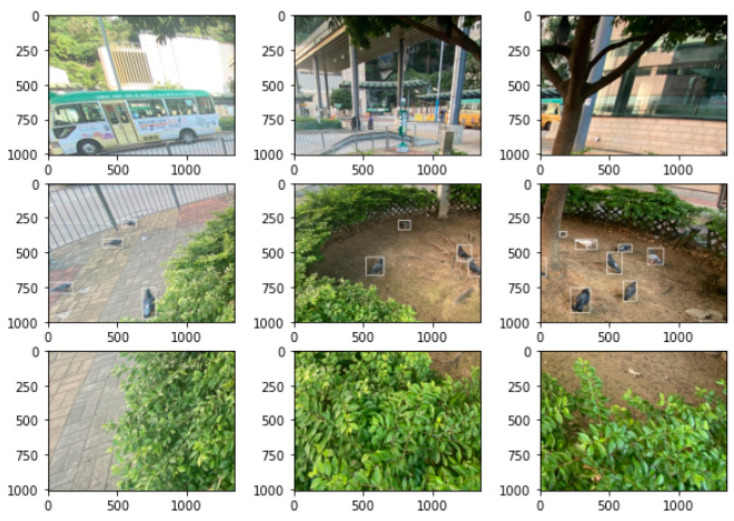
Images splitting with annotation box information.

**Figure 3 animals-14-00159-f003:**
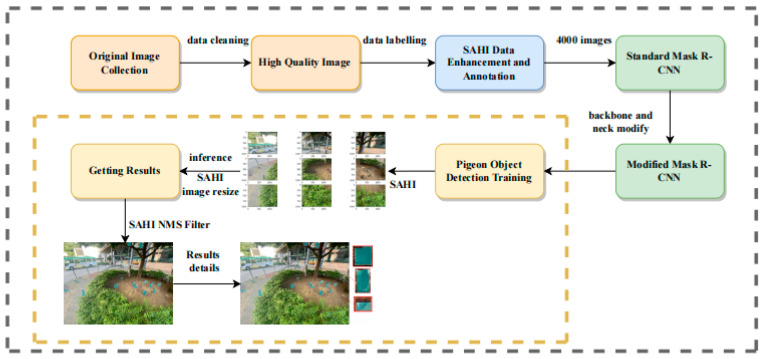
The pigeon detection system overview.

**Figure 4 animals-14-00159-f004:**
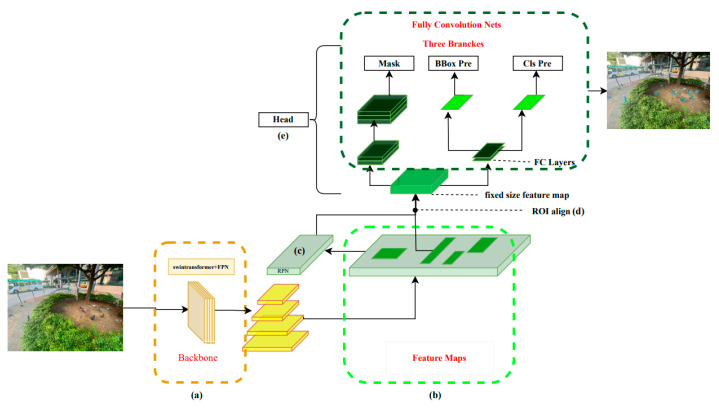
Swin-Mask R-CNN with SAHI model. a: Backbone, b: Feature Maps, c: Region Proposal Network (RPN), d: Region of Interest (ROI) Alignment, e: Head.

**Figure 5 animals-14-00159-f005:**
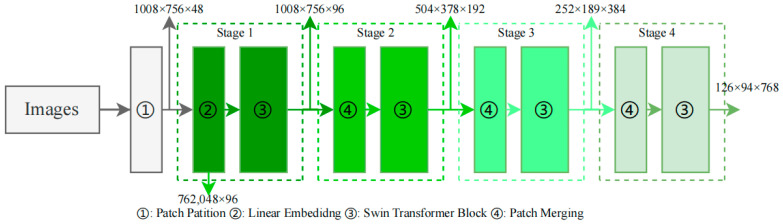
Swin Transformer backbone architecture detail.

**Figure 6 animals-14-00159-f006:**
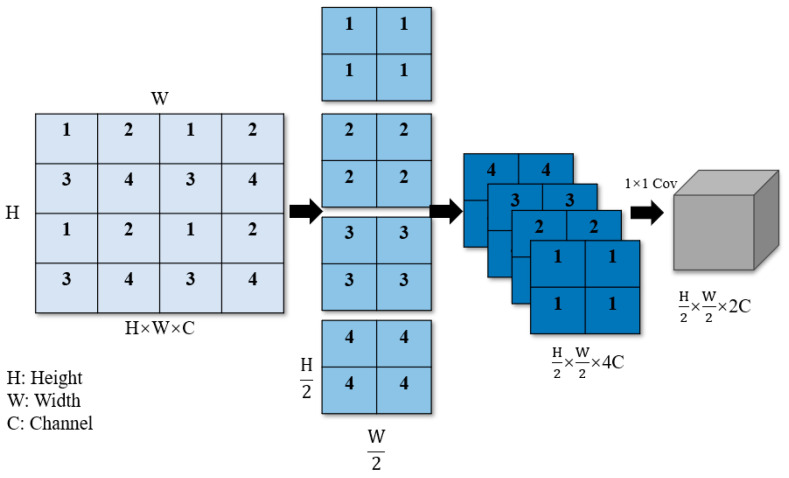
Patch merging in Swin Transformer.

**Figure 7 animals-14-00159-f007:**
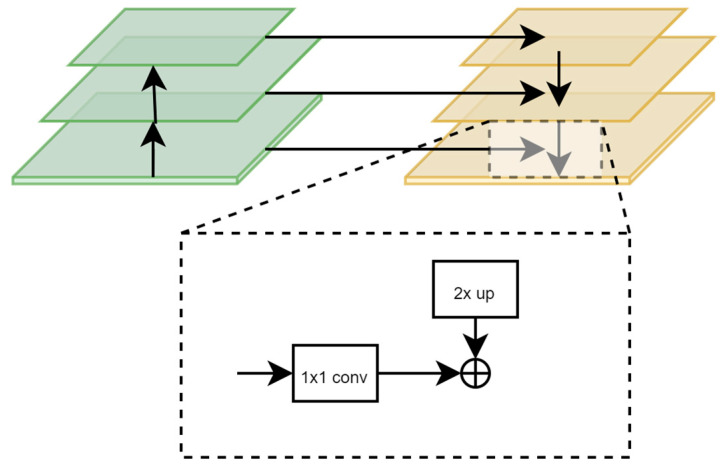
FPN Network.

**Figure 8 animals-14-00159-f008:**
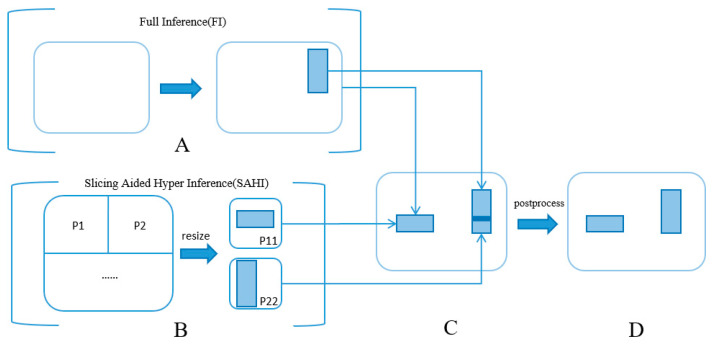
Slicing-Aided Hyper Inference (SAHI) process. A: Full Inference (FI), B: Slicing Aided Hyper Inference (SAHI), C: Initial Bounding Box results, D: the Final Result.

**Figure 9 animals-14-00159-f009:**
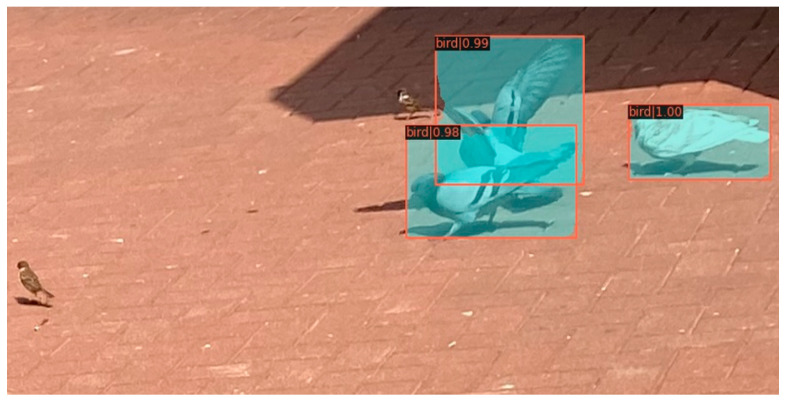
Results for predicting feral pigeon.

**Figure 10 animals-14-00159-f010:**
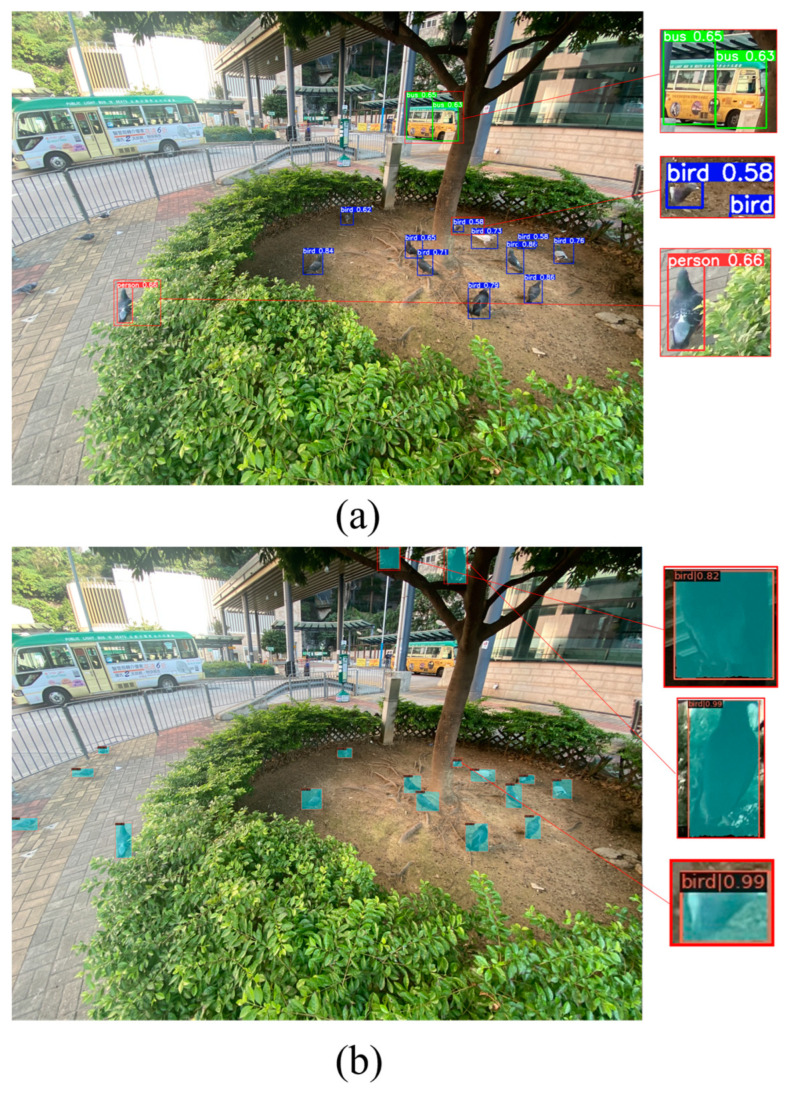
Results for prediction: (**a**) YOLOv5 in COCO80 classes; (**b**) Swin-Mask R-CNN with SAHI.

**Figure 11 animals-14-00159-f011:**
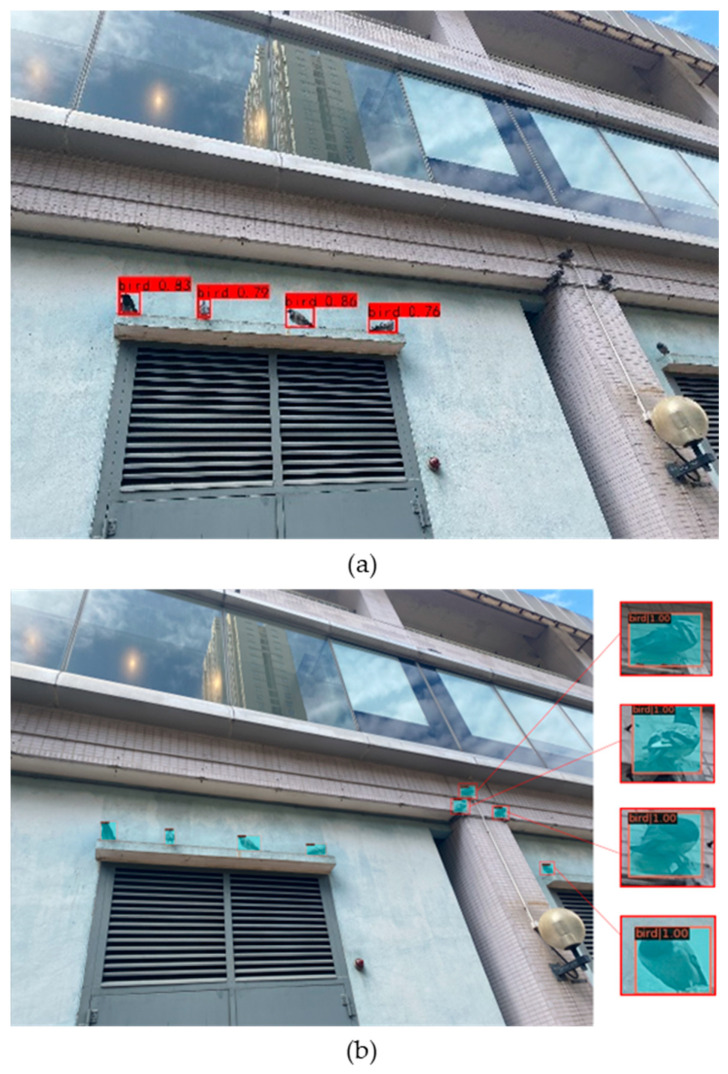
Results of prediction: (**a**) original Mask R-CNN; (**b**) Swin-Mask R-CNN Slicing-aided hyper inference after slicing-aided fine-tuning.

**Figure 12 animals-14-00159-f012:**
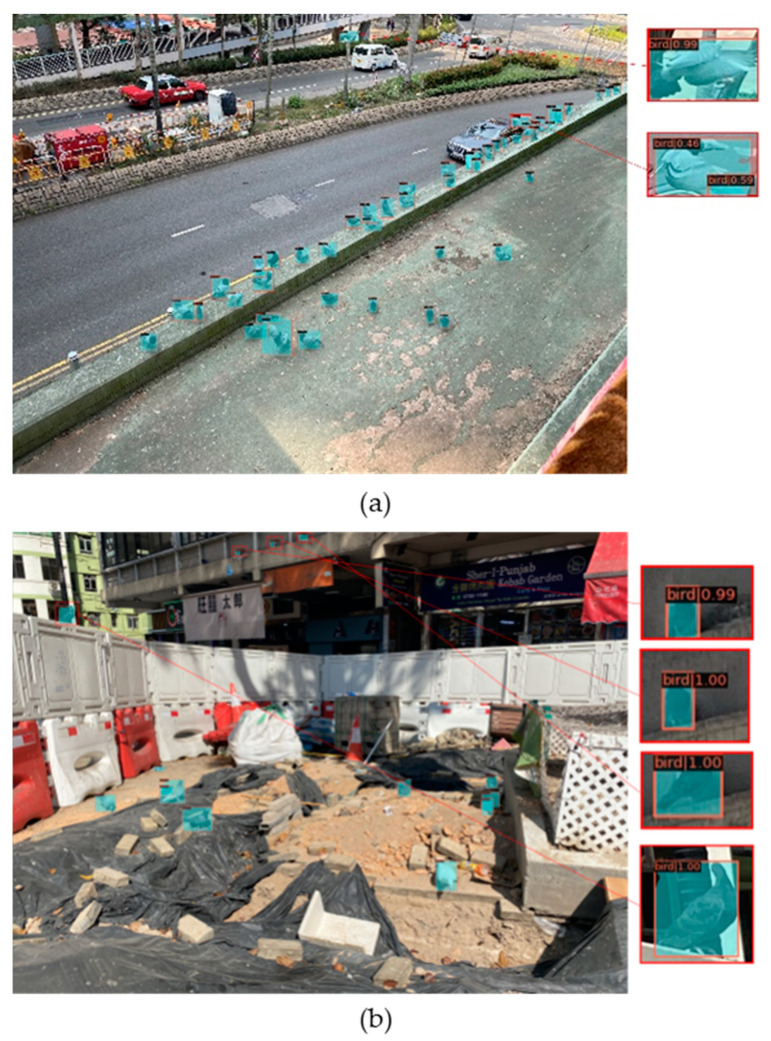
Feral pigeons with different postures in different backgrounds in the results in our model: (**a**) Road environment; (**b**) Roof environment.

**Figure 13 animals-14-00159-f013:**
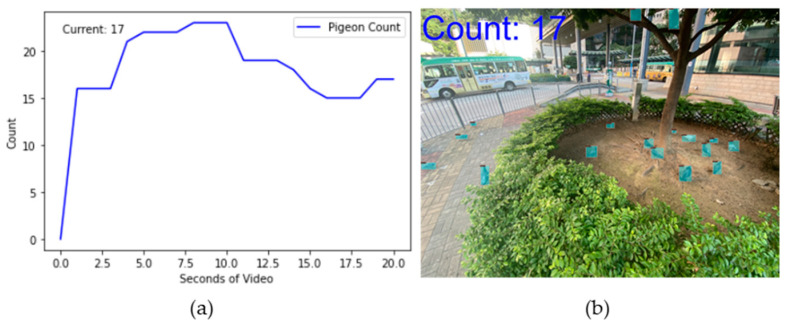
Dynamic data display of feral pigeon: (**a**) pigeon counting graph; (**b**) the image corresponding to the nth second.

**Table 1 animals-14-00159-t001:** Dataset splitting for modeling.

Dataset	Train	Validation	Test	All
Original dataset	266	67	67	400
Data augment	2400	600	600	3600
Final dataset	2666	667	667	4000

**Table 2 animals-14-00159-t002:** Experimental Environment and Model Evaluation Indicators.

Configuration	Parameters
CPU	32 vCPU AMD EPYC 7763 64-Core Processor
GPU	A100-SXM4-80GB (80 GB)
Development environment	Python 3.8
Operation system	Ubuntu 18.04
Operating Deep Learning Framework	Pytorch 1.9.0
CUDA Version	CUDA 11.1

**Table 3 animals-14-00159-t003:** The experiment of detection performance between different models.

Model	Backbone	Model Weight Size	mAP	AP_50_	AP_50s_
YOLOv5-s	Darknet53	72 m	0.45	0.65	0.36
YOLOv5-m	Darknet53	98 m	0.44	0.69	0.39
YOLOv5-s	GhostnetV2	73 m	0.42	0.62	0.33
YOLOv5-s	MobilenetV3	111 m	0.47	0.65	0.35
Faster R-CNN	Resnet	142 m	0.52	0.70	0.43
Mask R-CNN	Resnet	229 m	0.51	0.75	0.40
Mask R-CNN	MobileViT	240 m	0.61	0.77	0.47
Mask R-CNN	Swin Transformer	256 m	0.64	0.77	0.52
Modified Mask R-CNN (Swin-Mask-RCNN)	Swin Transformer + FPN	269 m	0.68	0.87	0.57

**Table 4 animals-14-00159-t004:** The ablation experiment of our model.

Step	Model	Backbone	Model Weight Size	mAP	AP_50_	AP_50s_
1	Mask R-CNN	Swin Transformer + FPN	269 m	0.68	0.87	0.57
2	Mask R-CNN	Swin Transformer	256 m	0.64	0.77	0.52
3	Mask R-CNN	Resnet	229 m	0.51	0.75	0.40

**Table 5 animals-14-00159-t005:** The experiment of detection performance between different models with SAHI.

Model	mAP	AP_50_	AP_50s_
YOLOv5-s + SAHI	0.51	0.71	0.42
YOLOv5-m + SAHI	0.56	0.74	0.46
YOLOv5-s + GhostnetV2 + SAHI	0.49	0.67	0.38
YOLOv5-s + MobilenetV3+ SAHI	0.53	0.70	0.41
Faster R-CNN + Resnet + SAHI	0.60	0.72	0.46
Mask R-CNN+ Resnet + SAHI	0.62	0.78	0.52
Mask R-CNN+ MobileViT + SAHI	0.69	0.83	0.62
Swin-Mask R-CNN + SAHI (ours)	0.74	0.93	0.67

**Table 6 animals-14-00159-t006:** The experiment of detection performance between different dataset sizes.

Model (Swin-Mask R-CNN + SAHI)	mAP	AP_50_	AP_50s_
400 images	0.51	0.71	0.42
2000 images	0.71	0.74	0.63
4000 images	0.74	0.93	0.67

## Data Availability

The data presented in this study are available on request from the corresponding author. The data are not publicly available due to the costs and resources involved in making the dataset publicly accessible.

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
