# Peer review of "Automatic Detection of Feral Pigeons in Urban Environments Using Deep Learning"

_animals, 2024, doi:10.3390/ani14010159_

Round 1
Reviewer 1 Report
Comments and Suggestions for Authors
The article proposes a deep learning model that utilizes computer vision technology to significantly improve the detection and population estimation of wild pigeons in Hong Kong's dynamic urban landscape. Its improved model, Swin Mask R-CNN with SAHI, integrates a Swin transformer network for deep feature extraction, a feature pyramid network for enhancing multi-scale learning, and three different detection heads for classification, bounding box prediction, and wild pigeon segmentation. With the help of the Slice Assisted Hyperinference Tool (SAHI), our model performs well in detecting small target pigeons in high-resolution images. The results indicate that compared to the mask R-CNN method, this improvement indicates that the success of our new method provides a promising solution for effectively managing urban wildlife populations. However, there are also some issues:
1. The abstract section of this article, although directly related to the topic, is relatively long and concise. The author needs to rewrite the abstract section and keep it between 150 and 200 words. (For details, please refer to and reference the " Spatial, Spectral, and Texture Aware Attention Network").
2. In the "Introduction" section, the author has written a lot of redundant content. For example, when introducing the deep learning model of this article, a brief introduction should be made, and the summary of the three key points is good. (For details, please refer to and reference the " Dual-branch collaborative learning network ").
3. The Fig2 image is somewhat blurry. Please provide a high-resolution and clear image. The background of the Fig3 flowchart, the color of the flowchart, and its layout need to be slightly adjusted.
4. The author of the discussion section should go straight to the topic, and the comparative experiment only compares with classical methods, lacking comparison with the latest methods. (Please refer to and refer to " Weighted Wavelet Visual Perception Fusion" for details).
5. Some of the references are older and should mostly cite references from recent years. (For details, please refer to and refer to " piecewise color correction and dual prior optimized contrast enhancement ").
Comments on the Quality of English LanguageThe article proposes a deep learning model that utilizes computer vision technology to significantly improve the detection and population estimation of wild pigeons in Hong Kong's dynamic urban landscape. Its improved model, Swin Mask R-CNN with SAHI, integrates a Swin transformer network for deep feature extraction, a feature pyramid network for enhancing multi-scale learning, and three different detection heads for classification, bounding box prediction, and wild pigeon segmentation. With the help of the Slice Assisted Hyperinference Tool (SAHI), our model performs well in detecting small target pigeons in high-resolution images. The results indicate that compared to the mask R-CNN method, this improvement indicates that the success of our new method provides a promising solution for effectively managing urban wildlife populations. However, there are also some issues:
1. The abstract section of this article, although directly related to the topic, is relatively long and concise. The author needs to rewrite the abstract section and keep it between 150 and 200 words. (For details, please refer to and reference the " Spatial, Spectral, and Texture Aware Attention Network").
2. In the "Introduction" section, the author has written a lot of redundant content. For example, when introducing the deep learning model of this article, a brief introduction should be made, and the summary of the three key points is good. (For details, please refer to and reference the " Dual-branch collaborative learning network ").
3. The Fig2 image is somewhat blurry. Please provide a high-resolution and clear image. The background of the Fig3 flowchart, the color of the flowchart, and its layout need to be slightly adjusted.
4. The author of the discussion section should go straight to the topic, and the comparative experiment only compares with classical methods, lacking comparison with the latest methods. (Please refer to and refer to " Weighted Wavelet Visual Perception Fusion" for details).
5. Some of the references are older and should mostly cite references from recent years. (For details, please refer to and refer to " piecewise color correction and dual prior optimized contrast enhancement ").
Author Response
Dear reviewer, thank you very much for taking the time to review this manuscript. Please find the detailed responses in the file we attached. And you also can find the modified contents we highlighted in the re-submit manuscript.

Reviewer 2 Report
Comments and Suggestions for Authors
Guo and team present a novel deep-learning model designed for detecting and estimating the population of feral pigeons within the dynamic urban environment. Leveraging a combination of cutting-edge technologies such as the Swin Transformer network, Mask R-CNN, and SAHI, the framework exhibits notable advancements in identifying small-target pigeons within high-resolution images. Despite its promising results, some revisions are crucial before publication.
The dataset recently compiled comprises 400 images, notably smaller than widely used datasets like COCO. The manuscript should explicitly address the dataset's limitations and explore how the model's performance might be affected with an increased volume of data.
The authors assert that the train/validation/test sets were randomly split; however, it remains unclear whether these subsets are drawn from the same distribution. Clarification is needed regarding potential similarities in scenes, resolutions, and the number/size of pigeons between the training and test sets.
Table 1 contains inaccuracies in the presented numbers. A thorough review and correction of the figures is warranted to ensure the accuracy of the reported results.
The choice of the Mask R-CNN as a benchmark for performance comparison appears outdated, given its publication over five years ago. The manuscript should include a discussion or comparison with more recent model developments and potentially assess the proposed model against state-of-the-art alternatives.
Given the dataset's limited size, concerns arise regarding the stability of the model's training. The author should furnish additional details on the model training process, including hyperparameter choices and convergence curve analysis, to provide a clearer understanding of the training dynamics.
Author Response

(The authors gave the same response as above.)

Reviewer 3 Report
Comments and Suggestions for Authors
The abstract discusses a deep learning model based on Mask R-CNN and Swin Transformer for monitoring feral pigeons in Hong Kong. The model uses a Slicing Aided Hyper Inference (SAHI) tool to enhance detection, particularly of small targets in large images, showing strong performance metrics, including a mean average precision (mAP) of 0.74 and improved recognition for small pigeons. However, there are some issues before publication.
1. In line 86-88, please add the paper "Yang X, Chai L, Bist R B, et al. A deep learning model for detecting cage-free hens on the litter floor[J]. Animals, 2022, 12(15): 1983." as a reference because it is a highly cited paper in bird detection and is also related to YOLO.
2. In line 178, please add more details about how you performed data augmentation and include relevant references.
3. In Figure 2, the top and bottom three pictures do not appear to contain any feral pigeons. Could you clarify whether these images are part of your dataset? Additionally, if your model is capable of detecting buses, please explain why there are no bounding boxes for buses, especially since you have included many for feral pigeons.
4. In line 218, please add a reference for SAHI.
5. Parts 2.5 to 2.6 are missing many references.
6. In the discussion section, apart from exploring further applications of the model, please add some discussion about how detecting feral pigeons can contribute to the ecosystem.
Author Response

(The authors gave the same response as above.)

Round 2
Reviewer 1 Report
Comments and Suggestions for Authors
This article proposes an improved deep learning model (swan mask R-CNN with SAHI) for wild pigeon detection. The model consists of three parts. Firstly, the SwinTransformer network (STN) extracts deep feature information. Secondly, the Feature Pyramid Network (FPN) integrates multi-scale features for learning at different scales. Finally, the three head branches of the model are responsible for classification, bounding box prediction, and segmentation. In the prediction stage, we use the Slice Assisted Hyperinference (SAHI) tool to focus on the feature information of small wild pigeon targets. Conduct experiments on the wild pigeon dataset to evaluate the performance of the model. The experimental results show that the model has good recognition performance for wild pigeons. However, there are also some issues:
1. The content of the first and second paragraphs of the data labels and enhancements in section 2.2 of the article is somewhat abrupt in terms of their vertical and horizontal connections.
2. The charts or flowcharts in sections 2.2 and 2.3 should provide a more detailed introduction to the experimental methods.
3. The scaled down flowchart in the bottom right corner of Figure 3 is not clear. It is recommended that the author make adjustments.
4. This article should add comparative experiments with the latest methods and refer to the latest methods of deep learning models. The author is expected to make adjustments to the comparative experimental methods section of the article.
5. The article lacks ablation experiments, and deep learning model formulas should be added as appropriate to better demonstrate the effectiveness of the experimental method in this article.
6. The dissertation section in the article is partially redundant and slightly lengthy. It is recommended that the author summarize it concisely.
Comments on the Quality of English LanguageThis article proposes an improved deep learning model (swan mask R-CNN with SAHI) for wild pigeon detection. The model consists of three parts. Firstly, the SwinTransformer network (STN) extracts deep feature information. Secondly, the Feature Pyramid Network (FPN) integrates multi-scale features for learning at different scales. Finally, the three head branches of the model are responsible for classification, bounding box prediction, and segmentation. In the prediction stage, we use the Slice Assisted Hyperinference (SAHI) tool to focus on the feature information of small wild pigeon targets. Conduct experiments on the wild pigeon dataset to evaluate the performance of the model. The experimental results show that the model has good recognition performance for wild pigeons. However, there are also some issues:
1. The content of the first and second paragraphs of the data labels and enhancements in section 2.2 of the article is somewhat abrupt in terms of their vertical and horizontal connections.
2. The charts or flowcharts in sections 2.2 and 2.3 should provide a more detailed introduction to the experimental methods.
3. The scaled down flowchart in the bottom right corner of Figure 3 is not clear. It is recommended that the author make adjustments.
4. This article should add comparative experiments with the latest methods and refer to the latest methods of deep learning models. The author is expected to make adjustments to the comparative experimental methods section of the article.
5. The article lacks ablation experiments, and deep learning model formulas should be added as appropriate to better demonstrate the effectiveness of the experimental method in this article.
6. The dissertation section in the article is partially redundant and slightly lengthy. It is recommended that the author summarize it concisely.
Author Response
Dear reviewer, thank you very much for taking the time to review this manuscript. Please find the detailed responses below and the corresponding revisions, and corrections highlighted in the re-submitted files. In the comparison experiment, we added the articles you recommended in the first round as references for the comparison test references.
- Zhang W, Zhou L, Zhuang P, et al. Underwater image enhancement via weighted wavelet visual perception fusion[J]. IEEE Transactions on Circuits and Systems for Video Technology, 2023.
- Zhang, Weidong, et al. Underwater image enhancement via piecewise color correction and dual prior optimized contrast enhancement. IEEE Signal Processing Letters 30, 2023, 229-233.

Reviewer 2 Report
Comments and Suggestions for Authors
The reviewer's questions have been answered, and the quality of the manuscript has been significantly improved after revision. Thus, I agree its publication in this journal.
Comments on the Quality of English LanguageMinor editing of English language required
Author Response
Dear reviewer, thank you very much for taking the time to review this manuscript. We did the minor editing of English language of this manuscript.